# Prevalence of lung cancer in Colombia and a new diagnostic algorithm using health administrative databases: A real-world evidence study

**Javier Amaya-Nieto**[1,2,3]☯*, **Gabriel Torres**[2]☯, **Giancarlo Buitrago**[1,2,3]☯

1 Department of Research and Innovation, Hospital Universitario Nacional de Colombia, Bogotá, Colombia,
2 Health Systems and Services Research Group, Universidad Nacional de Colombia, Bogotá, Colombia,
3 Instituto de Investigaciones Clínicas, Universidad Nacional de Colombia, Bogotá, Colombia

☯ These authors contributed equally to this work.
* jamayan@unal.edu.co

**Data Availability Statement:** Data cannot be shared publicly because of an confidentiality agreement between Universidad Nacional de Colombia and the Ministry of Health of Colombia.

## Abstract

Reliable, timely and detailed information on lung cancer prevalence, mortality and costs from middle-income countries is essential to policy design. Thus, we aimed to develop an electronic algorithm to identify lung cancer prevalent patients in Colombia by using administrative claims databases, as well as to estimate prevalence rates by age, sex and geographic region. We performed a cross-sectional study based on national claim databases in Colombia (*Base de datos de suficiencia de la Unidad de Pago por Capitación* and *Base de Datos Única de Afiliados*) to identify lung cancer prevalent patients in 2017, 2018 and 2019. Several algorithms based on the presence or absence of oncological procedures (chemotherapy, radiotherapy and surgery) and a minimum number of months that each individual had lung cancer ICD-10 codes were developed. After testing 16 algorithms, those with the closest prevalence rates to those rates reported by aggregated official sources (Global Cancer Observatory and *Cuenta de Alto Costo*) were selected. We estimated prevalence rates by age, sex and geographic region. Two algorithms were selected: i) one algorithm that was defined as the presence of ICD-10 codes for 4 months or more (the sensitive algorithm); and ii) one algorithm that was defined by adding the presence of at least one oncological procedure (the specific algorithm). The estimated prevalence rates per 100,000 inhabitants ranged between 11.14 and 18.05 for both, the contributory and subsidized regimes over years 2017, 2018 and 2019. These rates in the contributory regime were higher in women (15.43, 15.61 and 17.03 per 100,000 for years 2017, 2018 and 2019), over 65-years-old (63.45, 56.92 and 61.79 per 100,000 for years 2017, 2018 and 2019) who lived in Central, Bogota and Pacific regions. Selected algorithms showed similar aggregated prevalence estimations to those rates reported by official sources and allowed us to estimate prevalence rates in specific aging, regional and gender groups for Colombia by using national claims databases. These findings could be useful to identify clinical and economical outcomes related to lung cancer patients by using national individual-level databases.

Data are available from the integrated information system of the Ministry of Health of Colombia (https://www.sispro.gov.co/Pages/Home.aspx) for researchers who meet the criteria for access to confidential data.

**Funding:** This research was performed under a grant (grant number 420-2020) awarded to GB by The Colombian Ministry of Science, Technology and Innovation (Minciencias[https://minciencias.gov.co/]). The research project was also funded by AMGEN-Colombia (https://www.amgen.co/) under grant 627/2020. The funders had no role in study design, data collection and analysis, decision to publish, or preparation of the manuscript.

**Competing interests:** The authors have declared that no competing interests exist

## Introduction

Lung cancer has progressed to be one of the most frequent and deadly variations of cancer worldwide, even in low-middle-income countries (LMICs) [1]. Despite advances in diagnosing and treating this disease, lung cancer prevalence and mortality have increased. According to the last report of The Global Cancer Observatory (GLOBOCAN), this type of cancer represents the leading cause of mortality in cancer, with 23 deaths per 100,000 being reported globally [2, 3]. The low prevalence rate in LMIC is a concern because 80% of smokers live in these countries [4].

Health outcomes are worse in Latin American countries than in other regions because of their fragmented and underbudgeted health systems [5]. For optimal cancer care, LMICs must demonstrate significant prevention and early diagnosis strategies, as well as high quality measures, in medication, surgery and radiation delivery backed by patient support. However, health systems in LMICs are precarious, and most people with this condition are diagnosed in late stages if their cancer occurs, thus causing higher mortality rates [6].

Lung cancer in Colombia has an important impact, as it represents the fourth place in mortality among men and the fifth place among women [7]. Furthermore, a demographic change in the population has been described, which indicates an increase in individuals aged over 60-years-old [8]. It has also been described that lung cancer risk increases over time, especially in elderly people over 65-years-old [9, 10]. All of these combined factors can cause an increase in prevalence rates in some regions of the country due to the high prevalence of smokers or mining activities [11, 12].

The Colombian health system covers 99% of Colombian citizens [13]. It has a centralized fund controlled by the government that gathers payroll taxes and contributes an additional amount of money to ensure health care for the population, including the poorest individuals of this population. At the level of the insurers, there are public and private health institutions (known as *Empresas Promotoras de Salud* [*EPS*]) that act as intermediate payers between the centralized fund and health service providers (known as *Instituciones Prestadoras de Salud* [*IPS*]). These IPSs compete to be included in each EPS service network. Moreover, the insured population can be classified into a group that contributes to the health system via payroll taxes (known as the contributory regime) and a group funded by the national government (known as the subsidized regime) [14, 15].

This study aimed to develop an electronic algorithm to identify lung cancer-prevalent patients over 20-years-old in Colombia by using official databases, as well as to estimate prevalence rates of lung cancer by age, sex, and geographic region. This information could provide valuable information to Colombian policymakers, clinicians and the world to improve knowledge about lung cancer in LMICs.

## Methods

### Ethics and approval

This research was evaluated and approved by the ethics committee of the Universidad Nacional de Colombia as required by the national law [16]. This ethics committee waived the requirement of inform consent considering it was retrospective research and all data was anonymized in the datasets used for the study.

### Design and population

We performed a cross-sectional study based on national claims databases in Colombia (*Base de datos de suficiencia de la Unidad de Pago por Capitación* [UPC] and *Base de Datos Única de*

*Afiliados* [BDUA]) to identify lung cancer-prevalent patients in 2017, 2018 and 2019. These databases contain information on the population that was registered yearly from 2015 to 2019; additionally, these databases provide information used for this research under academic agreements between the Ministry of Health (MoH) and *Universidad Nacional de Colombia*. All the populations over 20-years-old were evaluated to identify individuals with lung cancer.

## Database characteristics

Several sources of information were used for this study. The first utilized source included the databases used by the Ministry of Health to calculate the sufficiency of the premiums that represents the amount of money that the government pays to insurers for each enrollee per year. It contained individual-level information about all the health services that were used by each person in the system reported yearly by each health provider. Data provided in this database include the location of each service that was used, the date of utilization, the type of health service, the diagnostic code (ICD-10) associated with each health service, the cost of the service, the health provider and the municipality. This information was available at an individual level and provided for the 10 largest insurers in the contributory regime that represent 88% of the contributory regime nationwide. In Colombia the contributory regime represents 45.8% (23.5 million people) of the population covered by the system. The system covers 99.06% of the population nationwide [17]. Information registered in the premium sufficiency databases was verified through a validation process designed by the Ministry of Health that involves correcting inconsistencies found in the data and the use of double validation with other databases used by the government. Information for the subsidized regime for Bogotá was also available by using the same information from the largest insurer in the city. The other database used includes sociodemographic and insurer (i.e., regime) information of all the enrollees in Colombia. These datasets were anonymized by the Ministry of Health before authorization of use was awarded to the Clinical Research Institute at Universidad Nacional de Colombia and they are further described in the supporting document (S1 File). In case datasets are required you can contact The Ministry of Health of Colombia using the official website: www.minsalud.gov.co or correo@minsalud.gov.co to get authorization for use.

Other secondary, aggregated and publicly available data sources were used. This information was collected from GLOBOCAN [18], National Cancer Institute [19] in Colombia and *Cuenta de Alto Costo* [20]. These data were used to verify the aggregated prevalence rates that were calculated in this study after having estimated differentiated prevalence rates over different age, region, and sex groups.

## Algorithms of case identification

To identify individuals over 20-years-old with lung cancer diagnoses, this study used 16 different electronic algorithms. Some of the algorithms were based on identifying any ICD-10 codes for tumors of the trachea, bronchi or lungs with malignant behavior (for example, roots C33 and C34) during a minimum number of months and at least one oncological procedure in the same individual in the previous three years (e.g., 2017, 2016 and 2015 for identifying people with lung cancer in 2018). We called these algorithms as "specific". The second type of algorithms were called as the "sensitive" algorithms by our team and aimed to identify patients by solely using the ICD-10 codes for tumors of the trachea, bronchi or lungs with malignant behavior during a minimum period of time. Specific and sensitive algorithms were compared, and two of them were chosen by selecting the closest prevalence rates to those rates reported by aggregated official sources (GLOBOCAN [18], National Cancer Institute [19] in Colombia and *Cuenta de Alto Costo-Colombia* [20]). Prevalence estimations were calculated for the

contributory by different strata of age, geographical region, and sex. For the subsidized regime this estimations were performed by different strata of age and sex.

## Prevalence method description

For global prevalence, the following equation summarizes the method that was used in this paper:

$$P_g = \frac{1}{N_{global}} \sum_{i=1}^{r} \sum_{j=1}^{q} \sum_{k=1}^{l} (n_{ijk} + m_{jk})$$

where $P_g$ is the general prevalence for Colombia for each year; $N_{global}$ is the total number of patients affiliated to the system by June 2017, June 2018 and June 2019 in those insurers aporting information to the premium estimation(*Unidad de Pago por Capitación*); $n_{ijk}$ is the number of prevalent cases identified for each region $i$, age group $j$ and sex $k$ in the contributory regime; and $m_{ijk}$ is the number of prevalent cases identified for each age group $j$ and sex $k$ in the subsidized regime.

## Results

After testing 16 algorithms, two electronic algorithms were selected, as they had the closest prevalence rates to those rates reported by aggregated official sources (GLOBOCAN [18], National Cancer Institute [19] in Colombia and *Cuenta de Alto Costo-Colombia* [20]). These comparisons were evaluated by comparing point estimates provided the mentioned sources to the confidence intervals calculated using bootstrapping with 1 000 subsamples as shown in Tables 1 and 2. The first selected algorithm was defined as the presence of ICD-10 lung cancer codes for four months or more (the sensitive algorithm), and the second selected algorithm was defined as the same criteria from the first algorithm and the presence of at least one onco-logical procedure (the specific algorithm).

By using previously selected algorithms, the identified general prevalence rates in the con-tributory regime were 11.15(IC 95 7.87 to 14.59), 11.14 (IC95 8.78 to 13.95) and 12.37(IC 95 9.41; 14.83) per 100,000 people with the use of the sensitive algorithm for the years 2017, 2018 and 2019, respectively. Whereas prevalence rates estimated using the specific algorithm were 7.09(IC 95 5.07 to 9.40), 7.37 (IC95 5.89 to 9.35) and 8.82(IC 95 6.94; 10.55) per 100,000 people for years 2017, 2018 and 2019 respectively (Table 1).

General prevalence rates in the subsidized regime for the years 2017, 2018 and 2019 were 14.75 (IC95 1.90 to 26.34), 18.05 (IC95 2.38 to 34.70) and 15.78 (IC95 1.32 to 24.54) per 100,000 people. Using the specific algorithm, general prevalence rates were 8.77 (IC95 0.95 to 13.88), 10.48 (IC95 0.95 to 17.30) and 10.39 (IC95 0.88 to 14.21), respectively. Distribution of estimations by sex and age groups is show in Table 2.

## Discussion

Using prevalence estimates is a useful strategy to design public policy and calculate attributable cost [21]. In this research two out of sixteen algorithms were selected as they had the closest prevalence rates to those rates reported by aggregated official sources. These algorithms were defined for this study as a sensitive algorithm and a specific algorithm because of their charac-teristics for identifying lung cancer cases, as described above. Using these two algorithms, the global prevalence rates for the contributory regime via the sensitive and specific algorithms ranged from 11.14 to 12.37 per 100,000 affiliates between 2017 and 2019 using the sensitive algorithm. In the case of the subsidized regime, general prevalence rates using the sensitive

**Table 1. Comparison between the prevalence rates identified in this study versus official aggregated data sources for 2017, 2018 and 2019 in the contributory regime.**

| | Contributory Regime | | | | | | Official sources | | |
|---|---|---|---|---|---|---|---|---|---|
| | 2017 | | 2018 | | 2019 | | | | |
| | Sensitive | Specific | Sensitive | Specific | Sensitive | Specific | CAC | NCI | GLOBOCAN |
| General prevalence | 11.15 (7.87; 14.59) | 7.09 (5.07; 9.40) | 11.14 (8.78; 13.95) | 7.37 (5.89; 9.35) | 12.37 (9.41; 14.83) | 8.82 (6.94; 10.55) | 10.5 | 9.8 | 15.7 |
| Sex | | | | | | | | | |
| Male | 11.95 (11.71; 12.17) | 8.26 (8.07; 8.44) | 11.54 (11.31; 11.76) | 8.19 (8.00; 8.37) | 13.16 (12.94; 13.39) | 9.72 (9.54; 9.89) | 10.1 | 12 | 16.9 |
| Female | 15.43 (15.18; 15.68) | 9.43 (9.24; 9.63) | 15.61 (15.35; 15.89) | 9.74 (9.54; 9.96) | 17.03 (16.75; 17.27) | 11.31 (11.10; 11.51) | 10.9 | 7.5 | 14.5 |
| Age group (years) | | | | | | | | | |
| 20–45 | 1.01 (0.95; 1.08) | 0.71 (0.65; 0.76) | 1.31 (1.24; 1.38) | 0.91 (0.85; 0.97) | 1.40 (1.33; 1.47) | 0.99 (0.93; 1.05) | 0.9 | N | 1.1 |
| 45–64 | 14.92 (14.57; 15.28) | 10.69 (10.39; 10.99) | 16.05 (15.71; 16.38) | 11.73 (11.43; 12.02) | 17.47 (17.12; 17.86) | 13.21 (12.91; 13.52) | 11.1 | N | 17.4 |
| > 65 | 63.450 (62.50; 64.37) | 38.64 (37.87; 39.39) | 56.92 (56.03; 57.80) | 35.24 (34.55; 35.94) | 61.79 (60.86; 62.66) | 41.04 (40.32; 41.78) | 49.2 | N | 82.7 |
| Region | | | | | | | | | |
| Atlantic | 7.90 (7.53; 8.25) | 5.12 (4.84; 5.41) | 8.87 (8.48; 9.27) | 6.06 (5.74; 6.38) | 9.42 (9.00; 9.82) | 6.97 (6.64; 7.31) | N | 7.2 | N |
| Bogota | 14.00 (13.66; 14.33) | 8.90 (8.63; 9.18) | 13.67 (13.35; 14.01) | 8.75 (8.48; 9.02) | 14.52 (14.19; 14.85) | 9.45 (9.19; 9.67) | N | 7.1 | N |
| Central | 18.99 (18.60; 19.40) | 12.47 (12.14; 12.79) | 18.79 (18.40; 19.17) | 12.36 (12.05; 12.67) | 21.84 (21.43; 22.24) | 15.20 (14.87; 15.54) | N | 16 | N |
| Oriental | 9.33 (8.97; 9.69) | 6.02 (5.73; 6.29) | 9.54 (9.17; 9.87) | 6.38 (6.08; 6.68) | 10.88 (10.51; 11.27) | 7.89 (7.55; 8.23) | N | 8.2 | N |
| Orinoquia/ Amazonia | 2.19 (1.45; 2.89) | 0.73 (0.36; 1.08) | 2.54 (1.81; 2.90) | 1.43 (0.73; 1.81) | 3.20 (2.51; 3.94) | 2.87 (2.15; 3.58) | N | 4 | N |
| Pacific | 14.47 (14.00; 14.90) | 9.30 (8.92; 9.68) | 13.41 (12.97; 13.87) | 9.26 (8.86; 9.62) | 14.33 (13.91; 14.79) | 10.53 (10.16; 10.92) | N | 9.4 | N |

Prevalence is shown in cases per 100,000

N: No information

NCI: National Cancer Institute

CAC: *Cuenta de Alto Costo*

GLOBOCAN: The Global Observatory of Cancer

algorithm were 14.75, 18.05 and 15.78 cases per 100,000 affiliates for years 2017, 2018 and 2019, respectively.

Estimation methods used by the first two institutions were based on data obtained from samples of a limited number of patients with lung cancer attended at various providers around the country, thus introducing possible bias to the calculated rates [22, 23]. Furthermore, prevalence rates reported by *Cuenta de Alto Costo* were based on data provided by insurers using clinical reports as data source; although it included information from 98% of the total population affiliated with the health system, these reported data could have underestimated the real prevalence, as it depended on passive reporting [24, 25]. In the case of our research, the identification process involved the use of algorithms that take information from the usage of the reported services registered in national administrative health claims databases. This represents an advantage in comparison to the other sources, as our estimates rely on service usage rather than an active process of reporting or an estimation performed over a limited sample.

In this study, the estimations obtained by the two algorithms offer some additional advantages for policy-makers and clinicians. First, it allowed us to calculate prevalence rates

**Table 2. Comparison between the prevalence rates identified in this study versus official aggregated data sources for 2017, 2018 and 2019 in the subsidized regime.**

| | Subsidized Regime | | | | | | Official sources | | |
| --- | --- | --- | --- | --- | --- | --- | --- | --- | --- |
| | 2017 | | 2018 | | 2019 | | | | |
| | Sensitive | Specific | Sensitive | Specific | Sensitive | Specific | CAC | NCI | GLOBOCAN |
| General prevalence | 14.75 (1.90; 26.34) | 8.77 (0.95; 13.88) | 18.05 (2.38; 34.70) | 10.48 (0.95; 17.30) | 15.78 (1.32; 24.54) | 10.39 (0.88; 14.21) | 10.5 | 9.8 | 15.7 |
| Sex | | | | | | | | | |
| Male | 9.38 (8.16; 10.28) | 5.78 (4.84; 6.65) | 10.92 (9.67; 12.01) | 6.84 (5.86; 7.62) | 10.96 (9.88; 12.07) | 7.70 (6.86; 8.78) | 10.1 | 12 | 16.9 |
| Female | 9.13 (8.19; 10.01) | 5.49 (4.78; 6.14) | 11.95 (10.92; 12.97) | 6.96 (6.14; 7.73) | 8.29 (7.46; 9.16) | 4.85 (4.26; 5.54) | 10.9 | 7.5 | 14.5 |
| Age group (years) | | | | | | | | | |
| 20–45 | 1.11 (0.81; 1.36) | 0.54 (0.27; 0.81) | 1.63 (1.07; 2.14) | 0.53 (0.27; 0.80) | 0.98 (0.74; 1.23) | 0.48 (0.25; 0.74) | 0.9 | N | 1.1 |
| 45–64 | 10.03 (8.90; 11.04) | 6.83 (5.70; 7.83) | 11.93 (10.59; 13.07) | 8.82 (7.77; 9.89) | 9.81 (8.50; 10.87) | 6.40 (5.44; 7.48) | 11.1 | N | 17.4 |
| > 65 | 32.05 (28.83; 35.42) | 18.16 (15.65; 20.59) | 40.25 (37.07; 43.52) | 21.73 (18.54; 24.18) | 34.64 (31.64; 37.67) | 22.63 (19.59; 24.86) | 49.2 | N | 82.7 |

Prevalence is shown in cases per 100,000

N: No information

NCI: National Cancer Institute

CAC: *Cuenta de Alto Costo*

GLOBOCAN: The Global Observatory of Cancer

differentiating between gender, regions and age groups. Second, it considered almost the total population affiliated with the contributory regime and the reported services that were utilized in the health system. Third, these estimations relied on the UPC administrative database, which represented consistent and validated data over the system, as it contains information with economical purposes. Fourth, it is possible to actively identify cases with lung cancer over the entire of population of patients attending to different providers. Finally, it is paramount to point out that these methods for identifying and estimating prevalence rates could help other LMICs to optimize their epidemiological surveillance system and to understand its impact in a context with different characteristics, compared to the developed world [26–29].

As LMIC countries enact changes to the population aging pattern, there is also a change in the burden of disease patterns from communicable to noncommunicable diseases (such as lung cancer), and efficient estimations are required to follow up on trends in public health [28, 30–32]. Authors such as Sierra et al. [33] have conducted research to address this problem and collected information from thirteen different countries in Central and South America. They found that lung cancer ranged from first to third regarding causes in incidence rates over these countries, thus showing concordance with a change in the aging patterns. In 2018, a study performed by Pilleron et al. [34] estimated that there were 679,000 new cases of cancer in Latin America and the Caribbean, and the most frequent variants were prostate, colorectal and lung cancer. While for 2020 the number of new cases estimated with lung cancer was 2,294,438 for the region according to GLOBOCAN and representing the second most frequent type of cancer [18]. The findings in this study showed a general prevalence in the contributory and the subsidized regimes ranging between 11.14 and 18.05 (using the sensitive algorithm) cases per 100,000 that positions the disease in the top ten list of prevalent cancer in the country and that corresponds to the prevalence that was identified in our study.

Another important aspect of the lung cancer prevalence rates is inequality and differences in its distribution. It has been observed that developed countries have increased prevalence

rates in comparison to developing countries [35]. There is also evidence showing that at the country scale, there is a trend of higher prevalence rates within lower socioeconomic groups. This situation can be explained by differences in smoking rates or other exposure factors, such as chemical exposure, mining activities or higher exposure to air pollution that usually affects lower income population groups or states within the country [36–39]. For this study, the results showed two important patterns related to his idea. The first result showed that there is a higher prevalence rate of the disease in regions with highly polluted cities or concentrations of mining activities. The second trend found that prevalence rates that were calculated in the subsidized regime were higher than in the contributory regime. This finding follows the literature and could be explained because of the inequalities in health care access that lead to worse healthcare conditions and there fore, more case of lung cancer being detected and treated. This ideas require more research in the future.

This study had several strengths and limitations. As a strength, this research used real-world evidence from administrative claim databases that were considered to be highly reliable regarding its quality and validation procedure. Information calculated via the sensitive and specific algorithms showed results that were similar to those reported by other institutions and a range of possible prevalence rates, rather than an absolute number. In contrast, there were some concerns regarding the estimation of prevalence rates in the subsidized regime because these data were calculated based on information provided by the largest insurer in the capital city; however, this is the best information available, and it could not be easily extrapolated to the country level. There is a natural challenge in estimating real prevalence rates of lung cancer via the selected algorithms because they identify only patients with the disease and are detected by the system. However, there is a proportion of patients with the disease that are not detected by the system; hence, they were not included in our estimation. When taking into account that health coverage in Colombia reaches 99% of the total population [13], we assume that this difference was very small. Finally, it is also possible that information bias affected the study; however, quality and validation processes that were implemented by MoH helped to control this risk.

## Conclusion

Selected algorithms showed similar prevalence estimations to those reported by official sources and allowed us to estimate prevalence rates in specific aging, regional and gender groups for Colombia by using national claims databases. These findings could be useful to formulate new public health strategies and to measure prevalence trends in lung cancer at a national scale by using administrative databases.

## Supporting information

**S1 File. Data sources description and Stata code used.**
(PDF)

## Acknowledgments

The authors thank Juan Camilo Forero and Juan Sebastian Castillo for their commentaries and support during the research process.

## Author Contributions

**Conceptualization:** Javier Amaya-Nieto, Giancarlo Buitrago.

**Data curation:** Javier Amaya-Nieto, Gabriel Torres.

**Formal analysis:** Javier Amaya-Nieto, Gabriel Torres, Giancarlo Buitrago.

**Funding acquisition:** Giancarlo Buitrago.

**Investigation:** Javier Amaya-Nieto, Gabriel Torres.

**Methodology:** Javier Amaya-Nieto, Gabriel Torres, Giancarlo Buitrago.

**Project administration:** Javier Amaya-Nieto, Giancarlo Buitrago.

**Software:** Javier Amaya-Nieto, Gabriel Torres.

**Supervision:** Giancarlo Buitrago.

**Validation:** Gabriel Torres.

**Visualization:** Javier Amaya-Nieto, Gabriel Torres.

**Writing – original draft:** Javier Amaya-Nieto, Gabriel Torres, Giancarlo Buitrago.

**Writing – review & editing:** Javier Amaya-Nieto, Gabriel Torres, Giancarlo Buitrago.

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
