## [Decision Letter · Decision Letter 0]

13 Sep 2022

PONE-D-22-13940Prevalence of Lung Cancer in Colombia and a New Diagnostic Algorithm Using Health Administrative Databases: A Real-World Evidence StudyPLOS ONE

Dear Dr. Amaya-Nieto,

Thank you for submitting your manuscript to PLOS ONE. After careful consideration, we feel that it has merit but does not fully meet PLOS ONE’s publication criteria as it currently stands. Therefore, we invite you to submit a revised version of the manuscript that addresses the points raised during the review process.

We look forward to receiving your revised manuscript.

Kind regards,

Raphael Mendonça Guimaraes, PhD

Academic Editor

PLOS ONE

Journal Requirements:

Reviewers' comments:

Reviewer's Responses to Questions

**Comments to the Author**

1. Is the manuscript technically sound, and do the data support the conclusions?

Reviewer #1: No

Reviewer #2: Partly

2. Has the statistical analysis been performed appropriately and rigorously? 

Reviewer #1: No

Reviewer #2: No

3. Have the authors made all data underlying the findings in their manuscript fully available?

Reviewer #1: No

Reviewer #2: No

4. Is the manuscript presented in an intelligible fashion and written in standard English?

Reviewer #1: Yes

Reviewer #2: Yes

5. Review Comments to the Author

Reviewer #1: The authors used an electronic algorithm to identify lung cancer prevalent patients in Colombia using administrative claims databases, as well as to estimate prevalence rates by age, sex, and geographic region. The topic is relevant, and the manuscript is well written. However, there are some major and minor issues that must be addressed.

Reviewer #2: In my opinion, although the study was properly designed, an important question of statistical analysis must be answered in order to support (or not) the authors' conclusion. Below are my comments.

The authors conclude: "selected algorithms showed similar prevalence estimations to those reported by official sources and allowed us to estimate prevalence rates in specific aging, regional and gender groups for Colombia by using national claims databases". The prevalence estimations by selected algorithms should be compared whit those reported by official sources by appropriate statistic test that would be able to test the differences between them. From that, the authors will can (or not) to sustain this conclusion.

Acronyms confuse the reader: they should be eliminated from the text.

6. PLOS authors have the option to publish the peer review history of their article (what does this mean?). If published, this will include your full peer review and any attached files.

Reviewer #1: No

Reviewer #2: **Yes: **Rafael Tavares Jomar

---

## [Author Response · Author response to Decision Letter 0]

20 Feb 2023

1. Academic Editor´s Comments / Raphael Mendonça Guimaraes, PhD

1.1. “Please ensure that your manuscript meets PLOS ONE's style requirements, including those for file naming.”

Answer: We reviewed links recommended and adjusted the documents we send as it complies with the requirements. 

1.2. “Please provide additional details regarding participant consent. In the ethics statement in the Methods and online submission information, please ensure that you have specified (1) whether consent was informed and (2) what type you obtained (for instance, written or verbal, and if verbal, how it was documented and witnessed). If your study included minors, state whether you obtained consent from parents or guardians. If the need for consent was waived by the ethics committee, please include this information.

If you are reporting a retrospective study of medical records or archived samples, please ensure that you have discussed whether all data were fully anonymized before you accessed them and/or whether the IRB or ethics committee waived the requirement for informed consent. If patients provided informed written consent to have data from their medical records used in research, please include this information.”

Answer: We complement ethics statement with clarifying information on inform consent. This study wasn’t required for inform consent by the ethics committee as it used data registered retrospectively and meant no risk for patients included in the research. All datasets used in the research and handled by the Ministry of Health of Colombia were anonymized before delivered to the researchers. This information was included in the body of the paper. 

1.3. “In your Data Availability statement, you have not specified where the minimal data set underlying the results described in your manuscript can be found. PLOS defines a study's minimal data set as the underlying data used to reach the conclusions drawn in the manuscript and any additional data required to replicate the reported study findings in their entirety. All PLOS journals require that the minimal data set be made fully available. For more information about our data policy, please see http://journals.plos.org/plosone/s/data-availability.

We will update your Data Availability statement to reflect the information you provide in your cover letter.”

Answer: individual level datasets used in this research were awarded to Professor Giancarlo Buitrago and the Clinical Research Institute at Universidad Nacional de Colombia for research purposes as it is supported by the documents “Annex 1. Authorization of handling datasets UNAL part1” and “Annex 2. Authorization of handling datasets UNAL part2”. For clarification about these statements, we included a footnote in the manuscript to explain authorizations given to the researchers and also contact information for those who want to ask the Ministry of Health of Colombia for permission to use the data. There were two sources of aggregated data that are cited in the methods of the document and for those we only used aggregated statistics available on the documents referenced. An additional description of the datasets used in the research was added in the supplement document as well. 

2. First Reviewer’s Comments

2.1. “The introduction is lengthy, and some sentences are repeated in the discussion and conclusion (e.g., last sentences, lines 81,82).”

Answer: we tried to state the objective of the research in the first paragraph of the discussion to guide readers over the results shown in this paragraph. However, as the recommendation of the reviewer, we removed these lines and modified them, so the paragraphs are shorter. 

2.2. “Despite the fact that the authors claim that the study was approved by an ethics committee, it is unclear what level of personal information was accessed (name, age, etc.). Please specify whether the information accessed was personal or non-identifiable. This point leads me to suspect that there is probably duplicated information. How many times can the same patient's information be entered into the database? Is there a unique identification number? Please elaborate on the strategies for removing repeat patients from the system. If none, acknowledge that over the limitations”

Answer:

a) This research used datasets with anonymized registries. All datasets used were anonymized as explained in the Ethics and approval section(it was added for this review). 

b) The Database characteristics section explains in detail datasets and variables included. It also explains that the Colombian Ministry of Health use a validation process to ensure information is valid and truthful (https://www.minsalud.gov.co/proteccionsocial/Paginas/rips.aspx). Because one of the datasets used corresponded to administrative claims, it was possible to for an anonymized ID to be duplicated in the initial database, however for the final analysis we used a dataset with the number of services utilized by each anonymized ID. This means that for the final dataset used for analysis we did not have duplicated registries as we could link all services from each individual to its anonymized ID. 

2.3. “Lines 87-94 - Please clarify the representativeness and coverage of the UPC and BDUA databases in relation to the entire Colombian population. It is also necessary to provide an estimated time for the system to register these medical claims.”

 Answer: Considering last information available for November 2022, contributory regime represents 45.8% of total population and 99.06% of the population is covered by the national system (meaning 99,05% belongs to the contributory regime or the subsidized regime). A footnote was added in the manuscript to explain this point. 

2.4. “Although the author stated in line 79 that the main goal of this article was to "...develop an electronic algorithm...", there is no information on how these algorithms were developed. The statistical methods used may be of interest to the readers; if they have already been published, please include references. If not, include them in the supplementary material, along with the .codes (.R code, Stata, Python, and so on). Please consider also including a description of the parameter used to fit these algorithms, methods, and, most importantly, the parameters measured to estimate the sensitivity and specificity, as well as the “gold standards”, include them either in the Methods or supplementary material.”

Answer: The manuscript explains in two subsections within the methods section the algorithms used for the analysis. The first one (Algorithms of case identification) explains that we used two kinds of algorithms (specific ones and sensitive ones). The specific one uses a combination of ICD-10 codes and procedure codes for radiotherapy, chemotherapy, or cancer surgery to identify cases. The sensitive one uses only ICD 10 for identification of cases. The second section (Prevalence method description) explains the formula used to calculate the prevalence in our study. This procedure basically adds up the number of cases identified by region, sex, and age group. We also explain in these two subsections that for selecting the final algorithms we compared our results to aggregated official sources using confidence intervals because official sources and available data does not provide enough information because they are not built with this purpose. We hope that existing prevalence reports improve over time and allow us to use a different approach. 

In regards of the code, we used and how we reached the estimations presented in the manuscript, we clarified at first that unfortunately data cannot be shared in order to reproduce the whole set of analyzes because of data privacy agreements between our University and the Ministry of Health. However, as we used Stata, we added a supplement to give access to the Stata code used in the analyzes, and, the log files that registered the estimation process. 

2.5. “Please explain why only 2017 estimates were provided. Consider providing prevalence estimates by year. It is also critical to include confidence intervals for your estimates, for example, Bootstrap techniques (talk with a colleague statistician to help with that).”

Answer: Prevalence estimations were only performed for 2017 because data from UPC was not fully updated by the time the manuscript was sent the first time. However, because of this comment we ask the ministry for an update in the data and added prevalence estimations for 2018 and 2019. These estimations represent the last information available as there is no updated data from 2020 to 2022 yet. 

2.6. “Globocan provides cancer prevalence estimates in 5-year, including all (diagnosed, in treatment, and cured), for example, the number of people alive today who were diagnosed with cancer in the previous five years. This fact poses some difficulties, and some comparisons made by the authors are invalid. Please, consider discussing if the dataset can identify when these patients were diagnosed with cancer for the first time. Furthermore, it is unclear whether these patients were alive or dead in 2017. Deaths should not be accounted for”

Answer: 

a) Comparisons in our research were done using a 3-year period available also at the GLOBOCAN website. The image added to this document show how this information can be consulted. (image added or data available at https://gco.iarc.fr/today/online-analysis-table?v=2020&mode=cancer&mode_population=continents&population=900&populations=900&key=asr&sex=0&cancer=39&type=2&statistic=5&prevalence=1&population_group=0&ages_group%5B%5D=0&ages_group%5B%5D=17&group_cancer=1&include_nmsc=0&include_nmsc_other=1). 

b) Because health claims data registers dates of usage, our study is able to identify when the patient comply with the definition explained in the manuscript. It also is able to exclude patients that were dead by the time of diagnosis because (i) We can cross-check data in these databases with vital statistics (we have individual level data of deaths using the same anonymized ID from RUAF) (ii) By definition subjects in our dataset must had utilized health services to be included by the definition, meaning they had to be alive by 2017. Finally, because of questions of data sources we added a supplement to the manuscript to further describe all the sources of information. 

2.7 “Because lung cancer is a common site of metastasis, the author may want to address this issue. Is it possible to tell which patients have primary site cancer and which have metastasis from another site? If not, how should this affect the projections?”

 Answer: This situation was considered at the beginning of the research by the team and because claim datasets do not include information about primary site of cancer, we decided to exclude all patients with other types of cancer from our estimates. 

2.8 “Some minor issues:

Line 179 – This sentence is redundant; consider removing it.

 Answer: Comment changed in the manuscript

Line 180 – Is there any evidence that these samples are chosen at random? If this is the case, the estimates should be unaffected.

 Answer: In this case we discussed between the authors and we believed this samples are not at random because there is a trend that for example, EPS (insurers) from rural areas are less likely to report information with enough quality (The Ministry of Health perform a validation process to data reported to the system). This situation may happen with other characteristics that do not allow the samples to be at random. 

Line 207 - Consider updating the estimates for 2020 (GLOBOCAN)

 Answer: This information was updated as recommended with the number of new cases for the year 2020 for the region according to information provided by GLOBOCAN. 

Line 218 - This sentence is unclear.”

 Answer: Sentence reviewed and corrected. 

3.Second Reviewer’s Comment / Rafael Tavares Jomar, PhD

3.1. “The authors conclude: "selected algorithms showed similar prevalence estimations to those reported by official sources and allowed us to estimate prevalence rates in specific aging, regional and gender groups for Colombia by using national claims databases". The prevalence estimations by selected algorithms should be compared whit those reported by official sources by appropriate statistic test that would be able to test the differences between them. From that, the authors will can (or not) to sustain this conclusion.”

 Answer: As reviewers pointed out, we reconsider our estimations and added confidence intervals for each parameter estimated. However, we also consider that as suggested by Zou et al. (1) confidence intervals can be used for identifying statistically significant differences in medicine estimations. This also can be interpreted as if one could replace hypothesis testing by this confidence interval approach. Another reason for not performing hypothesis testing on these aggregated official estimations is that their results usually do not present enough information for doing this kind of statistical testing as they are not built specifically for research. So, in this point we kindly ask the reviewers to take into account these reasoning. 

3.2. “Acronyms confuse the reader: they should be eliminated from the text.”

Answer: We tried to comply as much as we could to this recommendation. However, we kept some abbreviations in the text widely used in literature such as LMIC (low- and middle-income countries) or GLOBOCAN (Global observatory of cancer). The other ones were removed and used the full name of the institutions.

---

## [Editor Report · Decision Letter 1]

27 Feb 2023

Prevalence of Lung Cancer in Colombia and a New Diagnostic Algorithm Using Health Administrative Databases: A Real-World Evidence Study

PONE-D-22-13940R1

Dear Dr. Amaya-Nieto

We’re pleased to inform you that your manuscript has been judged scientifically suitable for publication and will be formally accepted for publication once it meets all outstanding technical requirements.

Kind regards,

Raphael Mendonça Guimaraes, PhD

Academic Editor

PLOS ONE
---

## [Editor Report · Acceptance letter]

2 Mar 2023

PONE-D-22-13940R1 

Prevalence of Lung Cancer in Colombia and a New Diagnostic Algorithm Using Health Administrative Databases: A Real-World Evidence Study 

Dear Dr. Amaya-Nieto:

I'm pleased to inform you that your manuscript has been deemed suitable for publication in PLOS ONE. Congratulations! Your manuscript is now with our production department. 

Kind regards, 

on behalf of

Dr. Raphael Mendonça Guimaraes 

Academic Editor

PLOS ONE